# Evolution of imprinting via lineage-specific insertion of retroviral promoters

Aaron B. Bogutz[1,5], Julie Brind'Amour [1,5], Hisato Kobayashi [2,5], Kristoffer N. Jensen[1], Kazuhiko Nakabayashi[3], Hiroo Imai[4], Matthew C. Lorincz [1]* & Louis Lefebvre [1]*

Imprinted genes are expressed from a single parental allele, with the other allele often silenced by DNA methylation (DNAme) established in the germline. While species-specific imprinted orthologues have been documented, the molecular mechanisms underlying the evolutionary switch from biallelic to imprinted expression are unknown. During mouse oogenesis, gametic differentially methylated regions (gDMRs) acquire DNAme in a transcription-guided manner. Here we show that oocyte transcription initiating in lineage-specific endogenous retroviruses (ERVs) is likely responsible for DNAme establishment at 4/6 mouse-specific and 17/110 human-specific imprinted gDMRs. The latter are divided into Catarrhini- or Hominoidea-specific gDMRs embedded within transcripts initiating in ERVs specific to these primate lineages. Strikingly, imprinting of the maternally methylated genes *Impact* and *Slc38a4* was lost in the offspring of female mice harboring deletions of the relevant murine-specific ERVs upstream of these genes. Our work reveals an evolutionary mechanism whereby maternally silenced genes arise from biallelically expressed progenitors.

[1] Department of Medical Genetics, Molecular Epigenetics Group, Life Sciences Institute, University of British Columbia, Vancouver, BC V6T 1Z3, Canada. [2] Department of Embryology, Nara Medical University, Kashihara, Nara 634-8521, Japan. [3] Division of Developmental Genomics, Research Institute, National Center for Child Health and Development, Setagaya, Tokyo 157-8535, Japan. [4] Molecular Biology Section, Department of Cellular and Molecular Biology, Primate Research Institute, Kyoto University, Inuyama, Aichi 484-8506, Japan. [5] These authors contributed equally: Aaron B. Bogutz, Julie Brind'Amour, Hisato Kobayashi. *email: matthew.lorincz@ubc.ca; louis.lefebvre@ubc.ca

Duentric regions acquire two characteristic epigenetic marks: transcription-coupled trimethylation at lysine 36 of histone H3 (H3K36me3)[1] and DNAme[2–5]. In both human and mouse female germline, the overall levels of CpG methylation increase from less than 5% in post-migratory primordial germ cells (PGCs) to ~40–50% in fully-grown, germinal vesicle oocytes (GVOs)[4,6–9]. In growing murine oocytes, 85–90% of this DNAme is deposited over H3K36me3-marked transcribed regions of the genome[3]. We recently demonstrated that SETD2-dependent deposition of H3K36me3 is required for de novo DNAme of transcribed gene bodies in mouse oocytes[10]. Such transcription-coupled de novo DNAme is also responsible for the establishment of regions of differential DNAme between the mature gametes (gametic differentially methylated regions, gDMRs), a subset of which are maintained throughout preimplantation development. This unique class of gDMRs, the imprinted gDMRs (igDMRs), can direct imprinted paternal allele-specific expression of the gene (s) under their regulation, the imprinted genes[3,11,12]. Importantly, transcription initiating within upstream oocyte-specific promoters has been reported to play a critical role in de novo DNAme at the maternal igDMRs of a number of mouse imprinted genes[3,13–17], but the evolutionary origin of such promoters remains unexplored.

Specific families of ERVs, also known as long terminal repeat (LTR) retrotransposons, are highly transcribed in mouse oocytes[3,18]. Such LTRs function as oocyte-specific promoters for both novel protein-coding genes and non-coding transcripts, the latter of which are widely distributed in intergenic regions[3,18,19]. We recently identified numerous LTR-initiated transcription units (LITs) in mouse oocytes associated with H3K36me3-coupled de novo DNAme[20]. Notably, CpG islands (CGIs) embedded within such LITs also become hypermethylated by this mechanism in oocytes[20] and the majority of maternal igDMRs associated with paternally expressed genes in both mice and humans overlap with CGIs[15,21]. Since ERVs are highly variable among mammalian species and de novo DNAme at maternal igDMRs is transcription-coupled, we hypothesized that active LTRs and their associated LITs may have played an important role in the genesis of lineage-specific maternal imprinting in mammals.

Here, we identify primate and rodent-specific igDMRs that appear to be de novo DNA methylated in oocytes as a consequence of transcription initiating within nearby LTR promoters. Our analysis of data from macaque and chimpanzee identifies Catarrhini- or Hominoidea-specific igDMRs embedded in oocyte transcripts emanating from species-specific LTRs. We further validate this phenomenon in two mouse mutants carrying deletions of upstream LTR promoters at *Impact* and *Slc38a4*, both of which lead to loss of oocyte DNAme acquisition at the igDMR in mutant females and loss of imprinted expression in the offspring. Together, our data suggest a model in which species-specific imprinted genes emerge from biallelically expressed progenitors via the acquisition of novel LTR promoters active during oocyte growth.

## Results

**LITs and the establishment of species-specific imprints.** To investigate the contribution of LITs to maternal gametic imprinting in humans and mice, we first curated a list that includes well characterized and putative maternally methylated igDMRs[22–28] by interrogating published whole-genome bisulfite sequencing (WGBS) data from gametes, placenta and somatic tissues (detailed in Supplementary Data 1). In total, we identified 21 mouse and 125 human igDMRs that were previously validated

through Sanger bisulfite sequencing, pyrosequencing, or whole-genome bisulfite sequencing (WGBS, see Methods and Supplementary Data 2). Among these, 6 and 110 maternal igDMRs are restricted to the mouse or the human lineage, respectively, while only 15 are conserved between the two species (Fig. 1a). We next applied de novo transcriptome assembly[29] to published human[30] and mouse[20] oocyte RNA-sequencing (RNA-seq) datasets and identified the transcript(s) and transcription start site (TSS) likely responsible for transcription-coupled deposition of DNAme over 20/21 mouse and 90/125 human igDMRs (Supplementary Data 2 and Supplementary Fig. 1a; identified using LIONS or de novo transcript assembly, see Methods)[29,31,32]. Among these, four mouse and 17 human maternal igDMRs, all of which are specific to the mouse or human genome, respectively, are embedded within or immediately downstream of LITs (Fig. 1a, Supplementary Fig. 1a, b and Supplementary Data 2). Compared with all genomic CGIs, this represents a significant enrichment of LITs at igDMRs (mouse: 4/21 igDMRs vs 152/16023 CGIs, Chi-square $p = 1.17 \times 10^{-17}$; human: 17/125 igDMRs vs 70/31144 CGIs, Chi-square $p = 7.42 \times 10^{-219}$). Thus, the presence of a lineage-specific proximal LTR that initiates an oocyte transcript overlapping with a genic gDMR is associated exclusively with genes showing evidence of a species-specific igDMR.

Of the 17 LITs associated with human-specific igDMRs, 12 initiate within primate (Hominoidea or Catarrhini)-specific ERV families (Fig. 1a and Supplementary Fig. 1c). Moreover, the four LITs apparently responsible for transcription-coupled de novo DNAme of the mouse-specific igDMRs, namely at *retro-Coro1c* (also known as *2010001K21Rik* and *AK008011*), *Cdh15*[27], *Slc38a4* (also known as *Ata3*)[33,34] and *Impact*[35], all initiate in rodent-specific ERVs (Fig. 1a and Supplementary Fig. 1c). While several such lineage-specific LTR families (i.e., LTR12C/MER51E or MTC/RMER19B) are actively transcribed in oocytes[19,20], others are generally expressed at low levels (Supplementary Fig. 1d, e), indicating that igDMR-coupled LITs do not necessarily initiate from LTR families that are widely expressed in oocytes. As hypothesized, the igDMRs associated with lineage-specific LITs also show species-specific hypermethylation in oocytes, which is retained (>35% DNAme) in the blastocyst[9], the placenta[26] or at least one somatic tissue surveyed, indicating that these genomic regions are indeed likely to carry imprinted DNAme marks (Fig. 1b and Supplementary Data 2). In accordance with previous reports of preferential maintenance of maternal germline-derived DNAme in human placenta[22,23,25,26,36], most of the LIT-associated human igDMRs retain DNAme in purified cytotrophoblast (CT; cells isolated from first-trimester human placenta)[26] but are hypomethylated in adult tissues (Fig. 1b). The remaining 5/17 human-specific maternal igDMRs (at *HTR5A*, *AGBL3*, *CLDN23*, *ZC3H12C*, and *SVOPL*) are associated with LITs driven by LTR families that colonized the common ancestor of the Euarchontoglires (rodents and primates, Fig. 1b). These specific LTR insertions, however, are not detected at the syntenic loci of the mouse genome and only 1/5 of the regions is methylated in mouse oocytes (the region syntenic to the human igDMR at *HTR5A*), which may be due to a distinct non-LTR-initiated transcript (Fig. 1b, Supplementary Fig. 1b and Supplementary Data 2).

This species-specific pattern of transcription initiation from an upstream LTR element leading to transcription-coupled establishment of a DNAme domain that includes the downstream igDMR/CGI can be visualized in a genome-browser view of orthologous regions (Fig. 1c, d). Note that we observe both sense and antisense configurations of the relevant LITs, driven by LTRs located either upstream or downstream (within an intron or 3′) of the regulated gene, respectively (Supplementary Fig. 2a). Of the 17 LITs putatively implicated in the induction of human igDMRs, 12 are in the 5′ sense configuration and five in an antisense

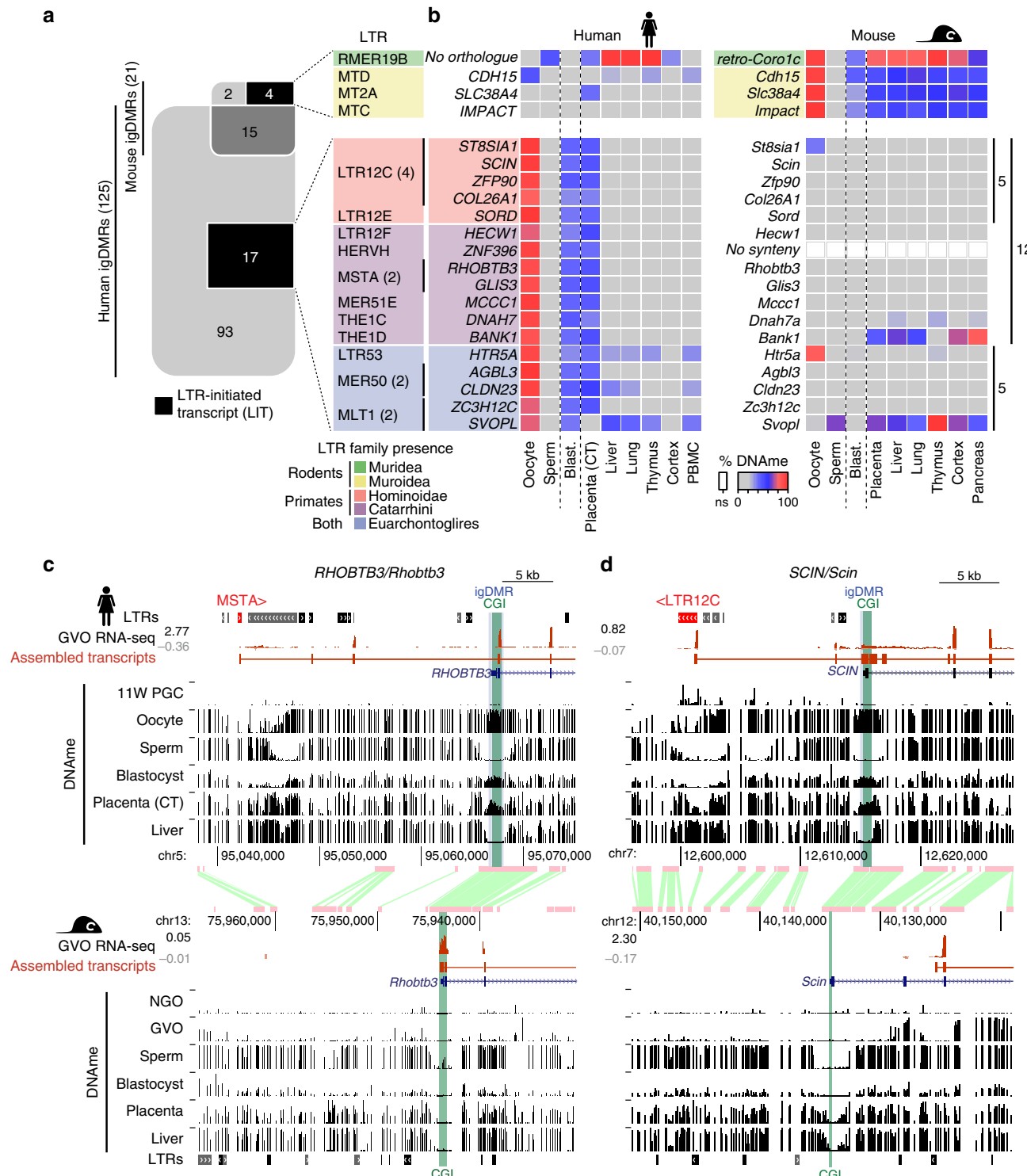

**Fig. 1 Identification of human and mouse maternal igDMRs embedded within lineage-specific LITs. a** Venn diagram showing the intersection of known maternal igDMRs in mouse and human, along with the subset of igDMRs in each species embedded within a LIT. For each LIT-associated igDMR, the family of the LTR in which transcription initiates in oocytes is shown on the right. The presence of each LTR family in relevant mammalian lineages is color-coded as in Supplementary Fig. 1c. **b** List of imprinted genes/igDMRs associated with LITs. Maternal igDMRs unique to mouse (4) or human (17) are shown, along with DNAme levels (heat map) for each igDMR in syntenic regions in the gametes, blastocyst, placenta and adult tissues in human and mouse. The retrogene *retro-Coro1c* is absent in the syntenic human region on chromosome 6p22.3 (*No orthologue*). *ZNF396* does not have a syntenic CGI in mice (*No synteny*). CT: cytotrophoblast; PBMC: peripheral blood mononuclear cell. **c, d** Screenshots of the human and mouse *RHOBTB3/Rhobtb3* and *SCIN/Scin* loci, including locations of annotated genes, LTR retrotransposons, and regions of syntenic homology. The relevant CGI, igDMR, and upstream LTR in human are highlighted in green, blue, and red respectively. For each species, RNA-seq data from GVOs are shown, along with assembled transcripts, including LITs and their 5′ LTR exons (red) for the human genes. DNAme levels in gametes, blastocyst, placenta, and liver are shown across each locus in both species. For the human DNAme data, profiles from female 11-week primordial germs cells are also shown (11W PGC) and oocyte DNAme is from a mixture of GVO and MII oocytes. Details of all the datasets used in this study are presented in Supplementary Data 1.

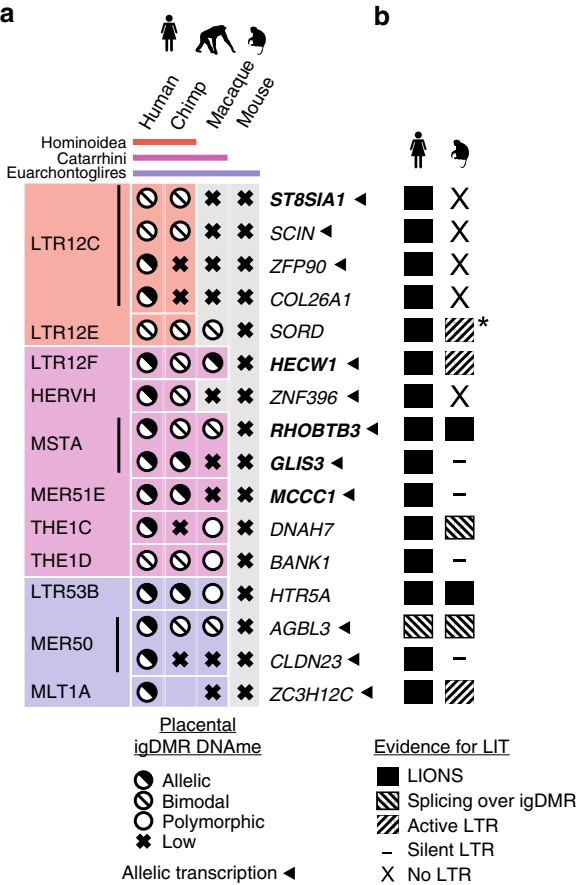

**Fig. 2 Conservation of oocyte LTR-initiated transcription and gametic imprinting in primates. a** Table of 16 human genes with igDMRs embedded within LITs active in oocytes and showing maternal/allelic DNAme in blastocyst and cytotrophoblast. The family of the initiating LTR is shown on the left, color-coded according to the phylogenetic distribution of the ERV family (top), as in Fig. 1a. For each species, the presence of the LTR insertion at each locus is indicated by a matching colored box and the igDMR DNAme status in human or the syntenic region in chimp, macaque and mouse placenta is shown. An empty box indicates no data. Arrows indicate genes for which evidence of allelic transcription has been published (Supplementary Data 2). Gene names in bold were analyzed in greater detail. **b** Conservation of LITs in human and macaque oocytes for the 16 igDMRs from panel **a**. Solid boxes indicate LITs discovered by LIONS, boxed hatches indicate LITs with evidence of splicing from the LTR over the igDMR, and unboxed hatches indicate evidence of transcription from the LTR. Dashes indicate LTRs from which no transcription is seen, and loci for which the relevant LTR is absent from the macaque genome are also shown (X: No LTR). Asterisk denotes an LTR12F that may initiate a LIT in macaque oocytes (Supplementary Fig. 4b).

configuration (Supplementary Table 1). At the *RHOBTB3/ Rhobtb3* locus for example, transcription in human oocytes initiates within an unmethylated primate-specific MSTA element located ~25 kb upstream of the promoter CGI/igDMR, forming a chimeric transcript that splices to the downstream genic exons of *RHOBTB3* (Fig. 1c). Coincident with this LIT, a large block of DNAme is deposited in oocytes over the promoter CGI and overlapping igDMR. Importantly, these regions are hypomethylated in human female 11-week gonadal PGCs[8] and in sperm. As previously documented for many human igDMRs[22,23,36], this imprint is maintained in the blastocyst and cytotrophoblast[26], but is hypomethylated (<2% DNAme) in adult tissues (Fig. 1b-c and Supplementary Data 2). Notably, *RHOBTB3* is expressed

predominantly from the paternal allele in human placenta[24,26,37,38]. In contrast, in mouse oocytes *Rhobtb3* transcription initiates at the promoter CGI, which is unmethylated in oocytes, placenta and adult tissues (Fig. 1b-c). Similarly, at the *SCIN* locus, a LIT initiates in an unmethylated LTR12C element ~14 kb upstream of the igDMR in human oocytes and extends into the gene, concomitant with de novo DNAme of this region between the PGC and mature oocyte stages (Fig. 1d). While the *SCIN* igDMR shows ~50% DNAme in human blastocyst and placenta (CT), the syntenic region in mice, including the *Scin* CGI promoter, is hypomethylated in each of these cell types and no upstream initiating transcript is observed in mouse oocytes (Fig. 1b, d). Consistent with the DNAme status of the locus in each species, *SCIN/Scin* is expressed exclusively from the paternal allele in human but not in mouse placenta[24,39]. Importantly, unlike in oocytes, eight of the nine genes that are associated with igDMRs and expressed in purified human cytotrophoblast (CT) show no evidence of transcription initiating from the proximal LTR in this cell type. Rather, for all six genes (*GLIS3*, *MCCC1*, *RHOBTB3*, *ZFP90*, *CLDN23*, *ZC3H12C*) that show clear paternally-biased expression (>70%) in CT (Supplementary Table 1), transcription initiates predominantly within the unmethylated igDMR promoter (Supplementary Fig. 2b, 3). Thus, imprinted expression of these genes in the placenta is correlated with LIT-associated deposition of DNAme over the igDMR in the oocyte and concomitant silencing of the maternal allele in the extraembryonic trophoblast lineage in the offspring.

**Primate LITs and de novo DNAme at maternal igDMRs.** Excluding the *SVOPL* igDMR, which is unmethylated in cytotrophoblast, we focused on the 16 human loci with evidence for maintenance of the maternal igDMRs in this placental lineage (Fig. 1b). Intriguingly, five of these, including *SCIN*, initiate in human ERV (HERV) families which colonized the common ancestor of the Hominoidea[40] (LTR12C or LTR12E), while seven of them initiate within families that colonized the primate lineage, including the common ancestors of the Catarrhini. The remaining four initiate in more ancient elements derived from LTR families common to both primates and rodents (Euarchontoglires) (Fig. 2a and Supplementary Fig. 1c). To characterize the relationship between LITs and the imprinting status of these gDMRs in non-human primates, we first determined whether the specific LTR insertions associated with these human placental igDMRs are annotated in the genomes of chimpanzees (*Pan troglodytes*; Hominoidea lineage) and rhesus macaques (*Macaca mulatta*; Catarrhini lineage). All 16 relevant LTR insertions are present in the chimpanzee genome. Moreover, all four insertions from LTR families that colonized the Euarchontoglires common ancestor and 6/7 LTR insertions from families that colonized the common ancestor of the Catarrhini are also present in the orthologous loci in macaque. Similarly, the Hominoidea-specific LTR12C and LTR12E families, which include LTRs driving transcripts overlapping the *ST8SIA1*, *SCIN*, *ZFP90*, *COL26A1*, and *SORD* igDMRs in human oocytes, are absent from the macaque genome (Fig. 2a and Supplementary Fig. 1c).

To assess the conservation of LITs and DNAme of associated gDMRs in macaque oocytes, we analyzed published RNA-sequencing[41] and genome-wide DNA methylome data[42]. As expected from the phylogeny of LTR12C elements, no LITs overlapping the region orthologous to the human igDMRs at the *ST8SIA1*, *SCIN*, or *COL26A1* loci were detected in macaque oocytes, and their CGIs remain hypomethylated (Figs. 2a, b, 3a and Supplementary Fig. 4a). Intriguingly, despite the absence of an LTR12E element, the region syntenic to the human *SORD* igDMR exhibits high levels of DNAme in macaque oocytes.

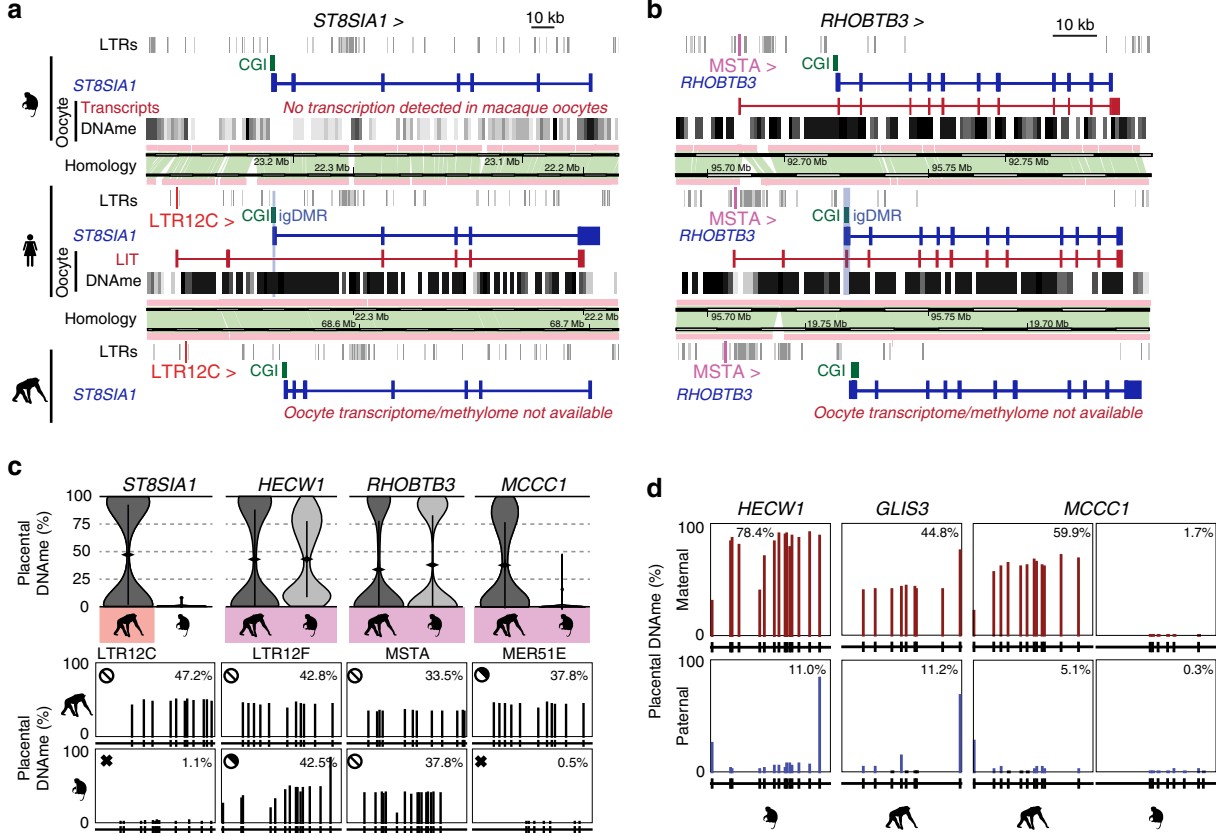

**Fig. 3 Chimp and macaque LITs and placental DNAme. a** Screenshot of the human *ST8SIA1* locus, showing its CGI promoter (green), annotated gene (blue), oocyte LIT (red), and DNAme in human and macaque oocytes (black). Highlighted are the location of the upstream LTR12C in human and chimp (red) and the human igDMR (blue). **b** Screenshot of the human *RHOBTB3* locus annotated as in **a** and highlighting the upstream MSTA LTR promoter (pink). **c** Violin plots of the distribution of mean DNAme levels per strand in placenta (chimp *MCCC1*: n = 2 all others: n = 3 biologically independent samples) for individual bisulphite-sequencing reads covering the igDMRs of the orthologous chimp and macaque *ST8SIA1*, *HECW1*, *RHOBTB3*, and *MCCC1* genes. Coloured boxes indicate the presence of the proximal LTR. For each gene, the mean DNAme level at each of the CpGs surveyed is shown below. Symbols for DNAme are as in Fig. 2a. Note that the MER51E in the macaque *MCCC1* locus is transcriptionally inert, likely due to macaque-specific SNPs rendering it transcriptionally inactive (Supplementary Fig. 4d–f). Source data are provided as a Source Data file. **d** Placental DNAme data at the *HECW1*, *GLIS3*, and *MCCC1* loci for informative samples heterozygous at SNPs of known parental origin, are shown for the species indicated.

Inspection of oocyte transcripts around the macaque *SORD* CGI reveals the presence of a highly transcribed LTR12F element oriented towards the putative igDMR, which may be responsible for deposition of DNAme at this locus (Supplementary Fig. 4a, b).

In contrast, LITs initiating in older LTR families that colonized the common ancestor of the Catarrhini or Euarchontoglire lineages show greater conservation between human and macaque oocytes. Indeed, 8/13 of such LTR insertions are also transcriptionally active in macaque oocytes, a subset of which display similar splicing events in both species (Fig. 2b). Remarkably, at least six of the associated CGIs are also hypermethylated in macaque oocytes (Supplementary Fig. 4a). For example, de novo methylation of the *RHOBTB3* CGI promoter/gDMR in both human and macaque oocytes appears to be the result of a conserved LIT initiating in a MSTA situated >20 kb upstream of the promoter, which clearly splices into the gene in both species (Fig. 3b). Other examples of LITs apparently conserved between the human and macaque lineages include those associated with the *HECW1* (LTR12F), *DNAH7* (THE1C), *HTR5A* (LTR53B), *AGBL3* (MER50), and *ZC3H12C* (MLT1A) putative igDMRs. In contrast, no LITs were detected at 4/11 loci despite the presence of a conserved LTR insertion in the orthologous macaque locus (Supplementary Figs. 4 and 5). For example, while the igDMR at the 5′ end of *GLIS3* is embedded within a LIT initiating in an active upstream MSTA in human oocytes, no LIT is detected in the orthologous region in the macaque locus nor is any RNA-seq coverage detected over the MSTA itself (Supplementary Fig. 5a, b).

The lineage-specific expression of orthologous LTRs may be explained by the accumulation over evolutionary time of indels and/or base substitutions that impact their promoter and/or splice donor activity. Indeed, sequence alignment of the orthologous MSTA insertions reveals a number of small deletions and single nucleotide polymorphisms (SNPs), which may impact transcription of the macaque LTR (Supplementary Fig. 5c). Similarly, despite the presence of a MER51E upstream of the *MCCC1* gene, no transcript initiating in this LTR is detected in macaque oocytes and no RNA-seq coverage is detected over the element itself (Supplementary Fig. 5d, e). Closer inspection of the MER51E insertions in the macaque locus reveals a number of SNPs and short INDELs relative to the orthologous MER51E in chimp and human, including mutations that likely disrupt an otherwise conserved PBX3 binding site that may render the LTR in the former inactive (Supplementary Fig. 5f). Taken together, these data indicate that the establishment of DNAme in oocytes at the igDMRs of a number of Hominoid- and primate-specific maternal igDMRs is likely induced by LITs originating in proximal lineage-specific LTR elements actively transcribed during oogenesis.

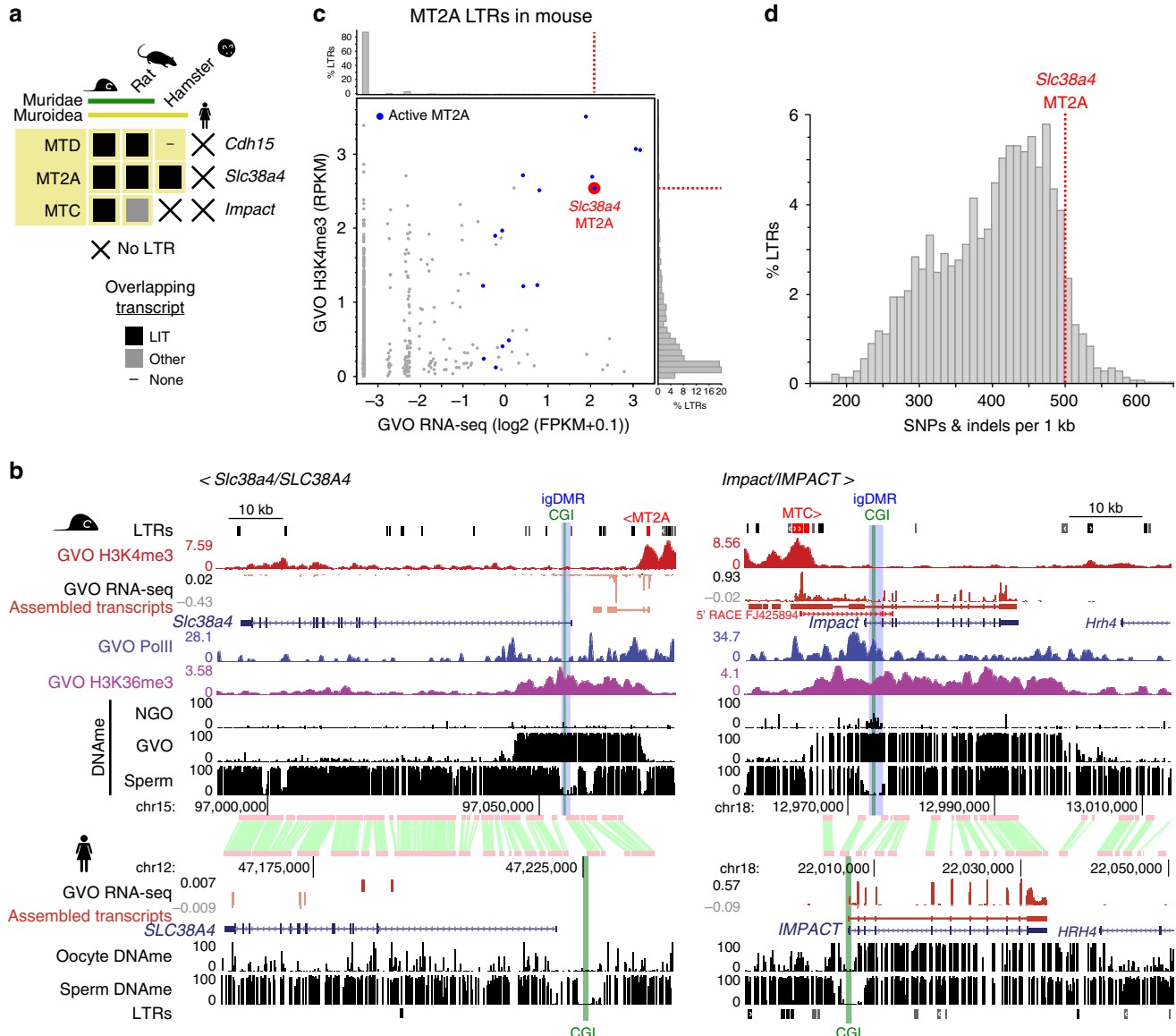

**Fig. 4 Phylogenetic relationship between LTR, LIT, and maternal DNAme at mouse-specific igDMRs. a** The presence or absence of a LIT overlapping the CGI at *Cdh15*, *Slc38a4*, and *Impact* are indicated with closed boxes and dashes, respectively. X: species in which the LTR is absent. **b** Screenshots of *Slc38a4/SLC38A4* and *Impact/IMPACT* loci showing GVO RNA-seq data (and the associated de novo Cufflinks transcript assembly), DNAme profiles in NGO, GVO and sperm, and H3K4me3, PolII, and H3K36me3 ChIP-seq tracks for mouse GVO. Highlighted are the upstream initiating LTR (red), CGI promoter (green), and mouse igDMR (blue). The syntenic region in human is also shown, including GVO RNA-seq and DNAme from oocytes and sperm. A 5′ RACE gene model initiating within the MTC element at the *Impact* locus is included in the right panel[14]. **c** Scatter plot of oocyte H3K4me3 and transcription levels for all mouse MT2A elements, with those acting as transcription start sites in oocytes highlighted in blue. The remaining transcribed elements reflect exonization events. The MT2A element initiating the LIT at *Slc38a4* (red) is amongst the most active elements of this family. **d** Histogram of the distribution of mouse MT2A LTRs as a function of divergence from the consensus sequence. The LTR driving expression at *Slc38a4* is amongst the most highly diverged.

**Conservation of LITs and igDMRs in apes versus monkeys**. To establish the imprinting status of these gDMRs in chimp and macaque placenta, we determined their methylation using a targeted high-throughput sodium bisulfite sequencing approach. We focused initially on the *ST8SIA1* gene, shown previously to be maternally methylated and paternally expressed in human placenta[24]. In chimpanzee, the genomic region syntenic to the human *ST8SIA1* igDMR is hypomethylated in adult tissues (Supplementary Fig. 4a), but shows a bimodal distribution of hypermethylated and hypomethylated sequenced reads in the placenta (Fig. 3c and Supplementary Fig. 6a), indicative of conserved placental-specific *ST8SIA1* imprinting within the Hominoidea. In contrast, the

syntenic CGI in macaque, as in mouse, is hypomethylated in the placenta (Figs. 2a and 3a, c). Therefore, the presence of an active LTR12C element ~40 kb upstream of the gDMR and its associated LIT are correlated with the imprinting status of the *ST8SIA1* promoter. *SCIN*, which is imprinted and paternally expressed in human placenta[24], also shows a bimodal distribution of DNAme in the orthologous region in chimp placenta. In contrast, in the absence of an LTR12C insertion or a LIT, the orthologous region in macaque placenta is hypomethylated (Supplementary Fig. 6a). On the other hand, despite the presence of a proximal LTR12C element in the chimp (as in the human genome), the regions syntenic to the human igDMRs at the *ZFP90* and *COL26A1* genes are

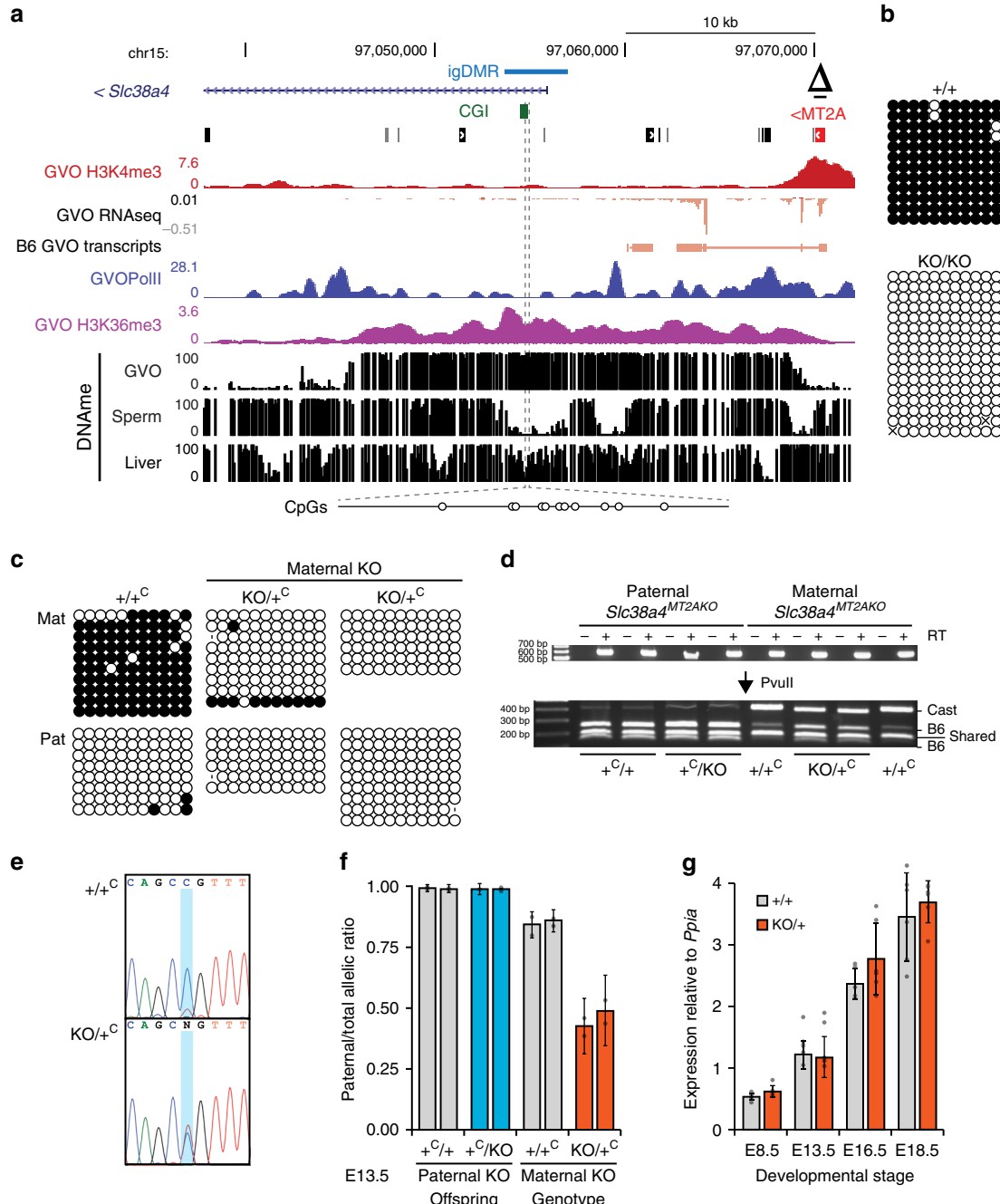

**Fig. 5 Loss of imprinting at *Slc38a4* upon maternal transmission of the MT2A KO allele. a** Genome-browser screenshot of the mouse *Slc38a4* promoter and upstream region, including the MT2A LTR (red), annotated *Slc38a4* exon 1, CGI (green), and igDMR (blue). GVO RNA-seq as well as RNA pol II, H3K4me3 and H3K36me3 ChIP-seq tracks are shown, along with DNAme data for GVO, sperm and adult liver. The region within the igDMR analyzed by sodium bisulfite sequencing (SBS), which includes 11 CpG sites, is shown at the bottom. Δ: extent of the MT2AKO deletion allele. **b** DNAme of the *Slc38a4* igDMR in GVO from wild-type and *Slc38a4*^MT2AKO/MT2AKO^ females determined by SBS. **c** DNAme of the *Slc38a4* igDMR in E13.5 (*Slc38a4*^+/MT2AKO^ × CAST)F1 embryos determined by SBS. Data for control (+/+^C^) and heterozygous (KO/+^C^) littermates with a maternally inherited MT2AKO are shown. +^C^: wild-type CAST allele; KO: *Slc38a4*^MT2AKO^. A polymorphic insertion in the amplified region allows for discrimination of maternal (Mat) and paternal (Pat) strands. Allele-specific expression analyses of F1 E13.5 placental RNA by **d** RT-PCR followed by PvuII RFLP analysis (RT reverse transcriptase), and **e, f** Sanger sequencing of a T ⟷ C transition in the 3′UTR of the *Slc38a4* cDNA (maternal B6: T allele; and paternal CAST: C allele). Each bar in **f** shows the mean of individual samples, and error bars show S.D. of two SNPs analyzed. Source data are provided as a Source Data file. **g** Total *Slc38a4* mRNA levels in E8.5, E13.5, E16.5, and E18.5 placentae, as determined by RT-qPCR (n = 6 biologically independent samples for each datapoint). Expression levels are relative to the housekeeping gene *Ppia*. Graph shows mean ± S.D. Source data are provided as a Source Data file.

unmethylated in chimp placenta (Fig. 2a), suggesting that, similar to the LTR insertions proximal to the *GLIS3* and *MCCC1* CGI in macaques (Supplementary Fig. 5), these orthologous LTR may be transcriptionally inert in chimp oocytes. This remains to be determined, however, as RNA-seq data from chimp oocytes is currently not available. The regions syntenic to the human *SORD* igDMR show bimodal DNAme in chimp as well as macaque, consistent with the presence of proximal active LTRs in human and

macaque oocytes. While the methylation of the macaque *SORD* locus is likely explained by the alternative LTR12F-initiated transcript described above (Supplementary Fig. 4b), it remains to be determined whether an antisense LTR12E-initiated LIT orthologous to that observed in human oocytes is responsible for depositing DNAme over the promoter CGI of this locus in chimpanzees.

As predicted by the presence of an orthologous LTR12F insertion oriented towards the *HECW1* gene in both the chimp and macaque loci, the genomic region of shared synteny to the human *HECW1* igDMR shows a clear bimodal distribution of hypermethylated and hypomethylated reads in the placentae of both species, indicative of conservation of LIT-associated *HECW1* imprinting in Catarrhines (Fig. 3c and Supplementary Fig. 4c). Moreover, exploiting a single SNP within the *HECW1* CGI, we observed clear allelic methylation (Fig. 3d and Supplementary Fig. 6b), confirming maternal allele-specific DNAme at the orthologous igDMR. Similarly, analysis of the chimp and macaque genomes in the regions syntenic to the *RHOBTB3* igDMR, which is embedded within an MSTA-initiated transcript in both human and macaque oocytes (Fig. 3b), reveals that the orthologous locus is also likely imprinted in these species (Fig. 3c). Furthermore, analysis of DNAme data from informative trios also reveals maternal-specific DNAme of the igDMRs at the *GLIS3* and *MCCC1* genes in the chimp placenta (Fig. 3c-d and Supplementary Fig. 6b). In agreement with the cognate LTR insertions being transcriptionally inactive in macaque oocytes (Supplementary Fig. 5), the genomic regions syntenic to the human and chimp igDMRs at both of these loci are hypomethylated in macaque placentae. Taken together, these observations reveal that transcription initiating within lineage-specific LTR elements in the oocytes of primates likely plays a critical role in the establishment of DNAme at proximal CGIs. The persistence of this oocyte-derived imprint in the placenta of their progeny yields maternal igDMRs that can potentially direct imprinted expression in this extraembryonic tissue, as shown for several human genes.

**Muroidea-specific LTRs and maternal igDMRs**. If the LTRs upstream of the mouse-specific igDMRs shown in Fig. 1 are indeed responsible for the establishment of maternal imprinting at these loci, then the orthologous genes should also be imprinted in those species harboring the same active LTR insertions. As the *retro-Coro1c* retrogene is absent from rats and more distantly related rodents (Supplementary Fig. 7a), we focused on the remaining three genes, namely *Cdh15*, *Slc38a4*, and *Impact* and their associated LTRs: MTD, MT2A, and MTC, respectively. All three LTR insertions are present in the orthologous regions of the rat (*Rattus norvegicus*) genome (Fig. 4a), consistent with the fact that these LTR families colonized the common ancestor of the Muroidea (Supplementary Fig. 1c, 7)[19]. In contrast, while the relevant MTD and MT2A elements are also present at the orthologous *Cdh15* and *Slc38a4* loci in the golden hamster (*Mesocricetus auratus*), the MTC element upstream of *Impact* is absent.

To assess whether the LITs identified at these imprinted loci in mice are also present in rat and golden hamster, we mined published oocyte RNA-seq data[19,20]. LITs emanating from the relevant LTRs were clearly detected at *Cdh15* and *Slc38a4* genes in rat oocytes, while in hamster, a LIT was detected only at the *Slc38a4* locus (Fig. 4a and Supplementary Fig. 7b–d). In rat, as in mouse[20], the *Slc38a4* LIT overlaps the DNAme block that extends into the 5′ end of the gene. In hamster, the MT2A-initiated LIT upstream of *Slc38a4* splices into exon 2 of the gene, creating a chimaeric transcript covering the entire gene, including the region syntenic to the mouse igDMR

(Supplementary Fig. 7c). In contrast, in human oocytes, the *SLC38A4* locus is not transcribed and the 5′ CGI is unmethylated (Fig. 4b). In the case of *Impact*, a LIT initiating in a co-oriented upstream MTC element encompasses the entire annotated gene in mouse and demarcates a hypermethylated domain, including the igDMR (Fig. 4b). Intriguingly, while the syntenic *Impact* CGI is hypermethylated in rat oocytes, the orthologous rat MTC insertion is apparently oriented on the opposite strand relative to the *Impact* gene (Supplementary Fig. 7d)[43]. Closer analysis of the locus reveals that a transcript encompassing the *Impact* CGI may originate from a distinct (non-LTR) upstream start site in rat oocytes, but a gap in the reference genomic sequence precludes detailed analysis of the upstream transcript at this locus (Supplementary Fig. 7d). Regardless, as in the mouse, *Impact* is expressed from the paternal allele in rat brain[43], consistent with epigenetic imprinting of its CGI and imprinted expression. Given the probable conservation of imprinting at *Slc38a4* and *Impact* in the Muroidea lineage, and the location of the intragenic igDMRs near their 5′ ends, we focused on these genes for further functional experiments in the mouse.

**An MT2A is required for *Slc38a4* imprinting**. In mouse oocytes, transcription over the *Slc38a4* igDMR originates in an H3K4me3 marked MT2A LTR located ~14 kb upstream of the gene, in a sense configuration. The 5′ end of the locus is embedded within the LIT, enriched for H3K36me3, and overlaps with a de novo DNAme block that encompasses the igDMR (Fig. 4b). In addition, our ChIP-seq analysis of RNA pol II in mouse GVOs reveals enrichment consistent with transcription extending from the MT2A through intron 1 of *Slc38a4* (Fig. 4b). MT2A elements are not generally transcribed at high levels in mouse oocytes, and analysis of all insertions >450 bp (1458 in total) indicates that only 16 initiate transcripts in GVOs at detectable levels (Fig. 4c and Supplementary Fig. 1e). As the MT2A upstream of *Slc38a4* has significantly diverged from the consensus sequence of this relatively old LTR family (Fig. 4d) and this LTR is not closely related to the few other MT2A elements active in oocytes (Supplementary Fig. 8), its transcriptional activity in the female germline may be due to the acquisition of novel transcription factor binding sites.

To test directly the role of the upstream MT2A insertion in *Slc38a4* imprinting, we generated a 638-bp deletion of the MT2A LTR in C57BL/6 N (B6) embryonic stem cells using CRISPR-Cas9 and flanking guide RNAs (Fig. 5a, Δ; Supplementary Fig. 9a). Germline transmission from male chimeras allowed us to establish the *Slc38a4*[MT2AKO] mutant mouse line on a pure B6 background. Heterozygous and homozygous animals were normal and fertile under standard husbandry conditions. In reciprocal crosses involving a wild-type and a heterozygous parent, heterozygotes were recovered at the expected frequency (Supplementary Table 2). We first assessed the effect of the MT2A deletion on DNAme at the *Slc38a4* igDMR in fully-grown oocytes using sodium bisulfite sequencing of 11 CpG sites within the igDMR located at the beginning of intron 1 (Fig. 5a). Whereas these sites are hypermethylated in wild-type oocytes, they remain unmethylated in oocytes from females homozygous for the MT2A deletion (Fig. 5b). To test for allele-specific transcription, we generated F1 progeny from reciprocal crosses with CAST/EiJ (CAST) mice, a hybrid strain background in which *Slc38a4* was previously shown to be imprinted[34]. Sodium bisulfite analysis confirmed that DNAme at the igDMR is absent in F1 embryos (E13.5) upon maternal transmission of the *Slc38a4*[MT2AKO] allele (Fig. 5c). Consistent with the absence of maternal DNAme, *Slc38a4* is expressed from both alleles in E13.5 placenta when the

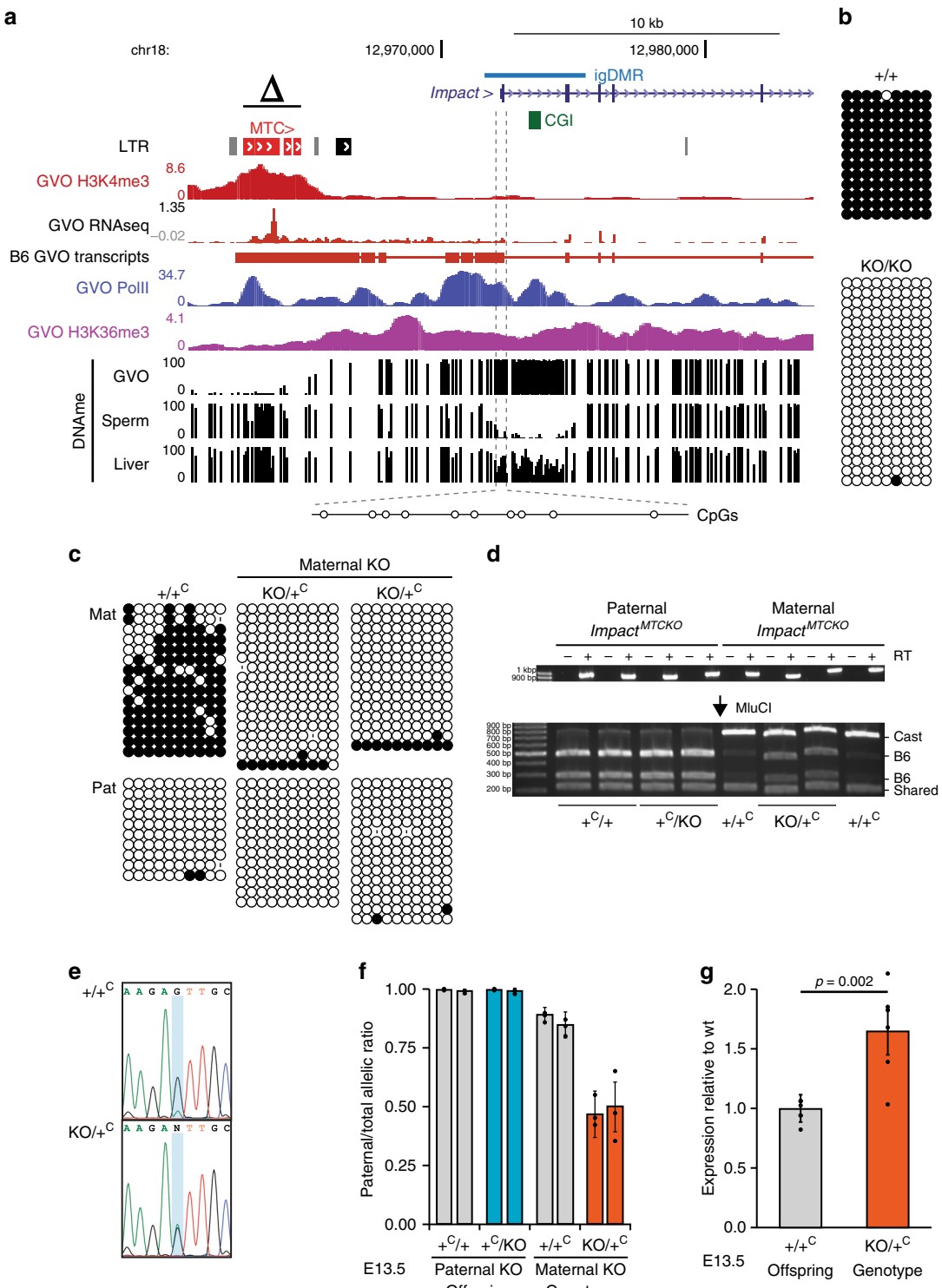

**Fig. 6 Loss of imprinting at *Impact* upon maternal transmission of the MTC KO allele. a** Genome-browser screenshot of the mouse *Impact* locus, including the upstream MTC LTR (red), CGI (green), and igDMR (blue). GVO RNA-seq as well as RNA pol II, H3K4me3, and H3K36me3 ChIP-seq tracks are shown, along with DNAme data for GVO, sperm, and adult liver. The region within the igDMR analyzed by sodium bisulfite sequencing (SBS), which includes 10 CpG sites, is shown at the bottom. Δ: extent of the upstream MTCKO deletion allele. **b** DNAme of the *Impact* igDMR in GVO from wild-type and *Impact^MTCKO/MTCKO* females determined by SBS. **c** DNAme of the *Impact* igDMR in E13.5 (*Impact^+/MTCKO* × CAST)F1 embryos determined by SBS. Data for control (+/+^C) and heterozygous (KO/+^C) littermates with a maternally inherited MTCKO are shown. +^C: wild-type CAST allele; KO: *Impact^MTCKO*. A polymorphic insertion and a SNP in the amplified region allow for discrimination of maternal (Mat) and paternal (Pat) strands. **d, e** Allele-specific expression analysis of F1 E13.5 embryonic head RNA by **d** RT-PCR followed by MluCI RFLP analysis, and **e** Sanger sequencing of a A ⟷ G transition in the 3′UTR of the *Impact* mRNA (maternal B6: A allele; and paternal CAST: G allele). RT: reverse transcriptase. Source data are provided as a Source Data file. **f** Quantification of relative levels of expression from the paternal *Impact* allele based on the analysis of embryos as in **e**. Graph shows mean ± S.D of three SNPs. Source data are provided as a Source Data file. **g** *Impact* mRNA levels analyzed by RT-qPCR on E13.5 embryonic RNA (*n* = 6 biologically independent samples). Expression levels are relative to those for the wild-type allele. Graph shows mean ± S.E.M. Source data are provided as a Source Data file.

deletion is maternally inherited, while monoallelic imprinted expression from the paternal allele is maintained in paternal heterozygotes (Fig. 5d-f). *Slc38a4* codes for an amino acid transporter (ATA3) highly expressed in the mouse placenta[34], suggesting that variations in its overall expression levels might lead to placental and/or growth abnormalities, as described for other imprinted genes. However, we did not observe either phenotype in maternal heterozygotes (Supplementary Fig. 9c–f). Notably, *Slc38a4* mRNA levels are high from E8.5 to E18.5 in wild-type placenta (Fig. 5g). However, consistent with the absence of placental or embryonic growth abnormalities, we found that loss of imprinting and biallelic expression at *Slc38a4* are not accompanied by an increase in total mRNA levels in maternal heterozygotes at those stages (Fig. 5g). These results, in accord with previously published findings[44], suggest that *Slc38a4* levels are normalized by transcriptional or post-transcriptional mechanisms in the placenta. Nevertheless, the possibility that a decrease in *Slc38a4* mRNA would bring its levels within the dosage-sensitive zone in which imprinting of the locus might provide a selective advantage is supported by the abnormal placental and embryonic growth phenotypes recently observed in *Slc38a4*-null conceptuses[45].

**An upstream MTC directs imprinting at *Impact*.** In mouse oocytes, the *Impact* gene is transcribed from an unmethylated upstream MTC element in the sense configuration. This LIT splices onto canonical exon 2 of *Impact*, as supported by RNA-seq data[3,20] and a 5′ RACE product from GVOs[15] (Fig. 4b). As seen at the *Slc38a4* locus, the MTC itself is marked by H3K4me3 in oocytes[46]. RNA pol II, H3K36me3, and DNAme are enriched over the transcribed region, which encompasses the entire *Impact* locus, including its igDMR and intronic CGI. Using a CRISPR-Cas9 mutagenesis approach similar to the one described for the MT2A at *Slc38a4*, we generated a mouse line in which a ~3 kb region upstream of the *Impact* gene, including the full-length MTC element, was deleted to generate the *Impact*[MTCKO] allele (Fig. 6a, **Δ**; Supplementary Fig. 9b). Following germline transmission through male chimeras, heterozygous males and females were bred to wild-type B6 mice and heterozygotes were recovered at the expected Mendelian ratios (Supplementary Table 2). Maternal and paternal heterozygotes, as well as homozygous mice of both sexes, appear normal and are fertile. As we observed for the MT2A deletion at *Slc38a4*, the igDMR at *Impact* fails to acquire de novo DNAme in mature oocytes from *Impact*[MTCKO/MTCKO] homozygous females (Fig. 6b).

To study the effect of the MTC deletion on imprinting of the downstream *Impact* gene, reciprocal crosses were performed between *Impact*[+/MTCKO] and CAST mice, and embryos were collected at E13.5. DNAme analysis of the *Impact* igDMR confirmed that DNAme is absent in embryos carrying a maternally inherited KO allele (*Impact*[MTCKO/+], Fig. 6c). Futhermore, allele-specific RT-PCR of head cDNA generated from the same E13.5 F1 progeny revealed that *Impact* is paternally expressed in all samples analyzed, with the exception of embryos in which the MTCKO allele is maternally inherited, where *Impact* is expressed from both parental alleles (Fig. 6d-f). Consistent with biallelic expression in these *Impact*[MTCKO/C+] embryos, quantitative analysis of total expression levels reveals that *Impact* mRNA levels are nearly doubled relative to those measured in wild-type controls (Fig. 6g). To determine whether this loss of imprinting at *Impact* influences the growth of mutant mice, we measured the weights of males and female wild-type and *Impact*[MTCKO/+]progeny. No significant weight difference was detected through postnatal week 60 (Supplementary Fig. 9g),

indicating that under standard husbandry conditions, increased *Impact* levels do not affect postnatal growth. Regardless, our analyses clearly reveal that DNAme and expression imprinting of the *Impact* gene in mice is dependent upon the presence of the upstream MTC element.

**Discussion**

Genomic imprinting is an epigenetic mechanism required for normal development and postnatal survival in mammals. Although more than a hundred imprinted genes have been identified and much has been learned about the epigenetic mechanisms regulating their monoallelic expression, it is still unknown how an ancestral gene, expressed from both alleles, acquires an imprinted expression pattern during evolution. Previous work has shown that through retrotransposition new retrogenes can acquire a maternal igDMR and show paternal allele-specific expression when inserted within a host gene expressed during oogenesis[47]. Although a number of such examples have been described in mammals[48], this phenomenon involves the creation of a new gene. Thus, it does not provide a model for the evolutionary switch from biallelic to imprinted expression manifest at species-specific imprinted genes, as explored in this study.

Here, we uncovered a novel mechanism whereby lineage-specific insertions of LTR retrotransposons, transcriptionally active during oocyte growth, can drive de novo DNAme and imprinting at a nearby gene. CRISPR-Cas9 mutagenesis at two mouse-specific imprinted genes, *Impact* and *Slc38a4*, confirmed the key role played by LTR-promoted transcription in guiding imprinting at these two mouse loci. Interestingly, *Slc38a4* was previously reported to be imprinted in mouse preimplantation embryos and early extraembryonic tissues in an H3K27me3-dependent manner[49]. However, the broad intragenic domain of H3K27me3 observed in oocytes and on the maternal *Slc38a4* allele in early embryos[49] is mutually exclusive of the upstream H3K36me3- and DNAme-marked domain studied here (Supplementary Fig. 10). This observation is consistent with a previous report showing that H3K36me3 inhibits PRC2 activity[50]. Thus, as with canonical imprinted genes, the *Slc38a4* igDMR in mice, which overlaps with an annotated TSS of this gene, is enriched for DNAme but devoid of H3K27me3. In addition to our results on the MT2A deletion allele, the importance of the LIT-induced igDMR for allelic usage is also supported by the observation of imprinted expression at *Slc38a4* in embryonic and adult epiblast-derived tissues in which the igDMR, but not the intragenic H3K27me3 domain, is present (Supplementary Fig. 10)[34,51]. The paradoxical critical role of two different maternal epigenetic marks laid down in two different regions of the *Slc38a4* locus may be reconciled by the fact that an alternative annotated TSS of the *Slc38a4* gene is located downstream of the igDMR and is embedded within the intragenic region enriched for H3K27me3 described by Inoue and colleagues[49]. This alternative TSS is used in adult liver, the tissue showing the highest *Slc38a4* expression levels after placenta (Supplementary Fig. 10). Although the igDMR is present in adult liver, *Slc38a4* expression is biallelic in this tissue, suggesting that tissue-specific loss of imprinted expression occurs via promoter switching[34]. Thus, depending on the tissue, imprinted expression of this gene likely requires one or the other epigenetic mark. Further studies are required to address this intriguing possibility and to determine whether establishment or maintenance of the igDMR at *Slc38a4* requires maternal H3K27me3. Notably, although preimplantation maintenance of the maternal DNAme imprint at *Impact* requires the protective action of the

KRAB-zinc finger proteins ZFP57 and ZFP445, the igDMR at *Slc38a4* does not[52]. This observation is consistent with a report showing that DNAme at the *Slc38a4* igDMR in postimplantation embryos is sensitive to loss of the H3K9 methyltransferase EHMT2/G9A[53].

Our observation that 15.5% (17/110) of human-specific and 66.7% (4/6) of mouse-specific maternal igDMRs are potentially induced by ERV promoters highlights the importance of this mechanism and its impact on the evolution of species-specific imprinted genes in mammals. Furthermore, our analysis also revealed evidence for convergent evolution of imprinting, where a distinct LTR may be responsible for imprinting in different species (LTR12E/F at the *SORD* locus in human/macaque for example). Further studies, such as those applied here in the mouse, will be required to determine whether alternative LTRs play a role in the imprinting of such loci in other species. Those species-specific maternal igDMRs not associated with a LIT are likely methylated as a consequence of transcription initiating in novel, single-copy, TSSs active in oocytes (as seen at the rat *Impact* locus).

While disruption of imprinting at some maternally silenced genes is associated with severe phenotypic outcomes, consistent with our observations at *Slc38a4* and *Impact*, loss of imprinting and biallelic expression of imprinted genes is not necessarily associated with obvious abnormal phenotypes. Previous studies of the paternally expressed genes *Plagl1*[3], *Peg3*[17], and *Zrsr1*[16] for example, revealed only subtle impacts on reproductive fitness following loss of imprinting, although detailed analyses of potential growth-related phenotypes have not always been reported.

While the role of active transcription in guiding DNAme of the oocyte genome, including at maternal igDMRs, is well established, the molecular mechanism involved still remains to be elucidated. We recently found that transcription-coupled deposition of H3K36me3 by SETD2 plays a critical role in this process, as DNAme at all maternal igDMRs, including at *Impact* and *Slc38a4*, is lost in mouse oocytes in which *Setd2* is deleted[10]. Thus, transcription-coupled H3K36me3 deposition is likely the critical common feature for the establishment of DNAme at maternal igDMRs in mammalian oocytes, including at igDMRs embedded within lineage-specific active LTR-initiated transcription units (Supplementary Fig. 1b).

A link between genomic imprinting and repetitive elements was explored previously by several groups, who proposed that the function of DNAme as a host defense mechanism might have been co-opted for allelic silencing at imprinted genes[54–57]. However, this hypothesis is inconsistent with our current understanding of de novo DNAme in the female germline, a process that is intimately associated with transcribed regions, which include imprinted gDMRs. Indeed, the mechanism we uncovered explains the establishment of DNAme imprints at single-copy CpG-rich regions that are unremarkable with respect to their repetitive element content. Rather, deposition of DNAme *in cis* is dependent upon the transcriptional activity of nearby ERVs that evade silencing in oocytes. In contrast, de novo DNA methylation at the *Rasgrf1* locus in male germ cells involves a *trans* mechanism, whereby an RMER4B LTR at the 3′-end of the gDMR is targeted for DNA methylation by a yet to be fully characterized nuclear piRNA pathway active in prospermatogonia[58]. Since the mechanism we described is not targeting repeat sequences, once the specific requirements for post-fertilization imprint maintenance are met, active ERVs, which propagate via retrotransposition, can theoretically induce imprinting of any downstream gene, with the prerequisite that the LTR be active in growing oocytes, when DNAme is established de novo by the DNMT3A/3L complex.

## Methods

**Data reporting**. No statistical methods were used to predetermine sample size. The experiments were not randomized and the investigators were not blinded to allocation during experiments and outcome assessment.

**Validation of mouse and human maternally methylated igDMRs**. To validate previously identified maternally methylated imprinted gDMRs (igDMRs)[22–25], we interrogated DNAme profiles from published whole-genome bisulfite sequencing data from gametes, placenta and somatic tissues (Supplementary Data 1)[59]. Specifically, we identified regions that are hypermethylated (>70% DNAme) in oocytes, hypomethylated (<30% DNAme) in sperm, retain DNAme (>25%) in the blastocyst, and show 35–65% DNAme in placenta (purified first-trimester cytotrophoblast; CT) or at least one adult somatic tissue[4,5,8,9,19,20,30,41,42,46,49,51,60–72]. We further validated and refined this list by only including gDMRs harboring either reported maternal, monoallelic or bimodal DNAme in the placenta and/or at least one somatic tissue. A bimodal DNAme pattern is a specific class of sequences methylated at ~50% for which individual DNA sequencing strands from WGBS datasets, although not overlapping a SNP with known parental origin, are either fully- (hyper) methylated or un- (hypo-) methylated. If the DNA sequence analyzed also encompasses a SNP, the bimodal pattern can be described as allelic when all hypo- or all hypermethylated strands contain the same variant at the SNP: only one of the parental alleles is methylated. Only when the parental origin of the sequence variants of this SNP is known in a given sample, can the allelic DMR be referred to as imprinted (maternally or paternally methylated). In total, 125 human igDMRs (46 of which are associated with reported paternal-biased transcription of the nearby gene) and 21 mouse igDMRs (all with reported paternal-biased transcription) are presented and analyzed in Supplementary Data 2. Syntenic regions between the mouse (mm10) and human (hg19), as well as chimpanzee (panTro4), macaque (rheMac8), rat (rn6), and golden hamster (mesAur1) genomes were obtained using the Liftover tool from the UCSC Genome Browser (http://genome.ucsc.edu/)[73].

**Identification of LTR transcripts overlapping igDMRs**. Oocyte RNA-seq libraries (see Supplementary Data 1) were aligned to the mm10 (mouse), hg19 (human), rhemac8 (macaque), mesAur1 (golden hamster), and rn6 (rat) assemblies using STAR v.2.4.0.i42[74]. De novo transcriptome assemblies were produced using Stringtie v.1.3.5 (mouse, human, golden hamster and rat)[32] or Cufflinks v.2.1.121 (macaque)[29] with default parameters. All de novo transcripts overlapping or in proximity of putative imprinted gDMRs were identified, and their transcription initiation site was confirmed by visual inspection of splice sites. Transcripts initiating in transposable elements were identified using LIONS[20,31]. The 5′ ends of de novo assembled transcripts were classified based on overlaps with UCSC Repeat Masker annotation and our putative igDMR list, and transcripts with the Up or UpEdge classification (per LIONS raw output) were taken into consideration. For manual inspection of LITs over specific CGI promoters, LITs were either identified by manually validating transcripts with EInside transposable element contribution in the LIONS raw output or by intersecting de novo transcripts with the boundaries of hypermethylated domains.

**Evolutionary tree of LTR families**. Phylogenetic tree of species shown was generated from TimeTree (http://timetreebeta.igem.temple.edu/). Integration of LTR families was imputed based on the presence or absence of the family in the species investigated, and colonization time is indicated at the root of those species in which it was found in common. For examining the evolutionary history of MT2A family, including the MT2A apparently responsible for de novo methylation of the *Slc38a4* igDMR in mouse oocytes, we identified all annotated (UCSC RepeatMasker) MT2A elements >450 bp (the MT2A consensus sequence is 533 bp). Multiple sequence analysis and phylogenetic tree construction were carried out with MEGA X[75] using MUSCLE UPMGA algorithm and by Neighbour-Joining method, respectively. Visualization and annotation of the phylogenetic tree were carried out with iTOL (https://itol.embl.de/)[76].

**Ethical approval for animal work**. All mouse experiments were approved by the UBC Animal Care Committee under certificates A15-0291 and A15-0181, and complied with the national Canadian Council on Animal Care guidelines for the ethical care and use of experimental animals. All primate experiments were approved by the animal experiment committee of Primate Research Institute (PRI) of Kyoto University (Approval No: 2018-004). Three rhesus macaque (Macaca mulatta) placentas and their parental blood DNA were collected and stored at −80 °C before use. Three chimpanzee (Pan troglodytes verus) placentas and their parental blood DNA were provided from Kumamoto Zoo and Kumamoto Sanctuary via Great Ape Information Network (GAIN).

**SNP genotyping and targeted bisulfite analysis in primates**. Regions orthologous to human placental igDMR were PCR-amplified with TaKaRa EX Taq HS (TaKaRa Bio) with primers specific for chimpanzee and rhesus macaque CGIs (Supplementary Data 3). PCR were performed under the following conditions: 1 min at 94 °C, then 40 cycles of 30 s at 94 °C, 30 s at 57 °C, 30 s at 72 °C, and 5 min final elongation at 72 °C. PCR products were purified with QIAquick PCR

Purification Kit (Qiagen). Sanger sequencing was carried out for each amplicon using the BigDye Terminator v3.1 Cycle Sequencing Kit and the ABI 3130xl DNA Analyzer (Thermo Fisher Scientific). The sequencing data were assembled using ATGC sequence assembly software (bundled with GENETYX Ver. 14, Genetyx Corporation) to identify SNPs at a given loci and determine the genotypes for each individual sample. Genomic DNA (100 ng per sample) was treated with sodium bisulfite using the DNA Methylation Gold Kit (Zymo Research). The bisulfite-treated DNA was PCR-amplified with TaKaRa EpiTaq HS (TaKaRa Bio) with primers specific for chimpanzee and rhesus macaque CGIs (Supplementary Data 3). PCR were performed under the following conditions: 1 min at 94 °C, then 35–37 cycles of 30 s at 94 °C, 30 s at 57 °C, 30 s at 72 °C, and 5 min final elongation at 72 °C. PCR products were purified with QIAquick PCR Purification Kit (Qiagen). Amplicons were tagged using NEB Next Ultra II DNA Library Prep Kit for Illumina (NEB) with five thermal cycles for amplification and subjected to paired-end sequencing on a MiSeq platform using MiSeq Reagent Nano Kit v2, 300 Cycles (Illumina). We used the QUMA website (http://quma.cdb.riken.jp/top/index.html) to quantify CpG methylation levels of each CGI. From each sequence data (fastq files), the top 400 forward reads (R1) were extracted and mapped to each CGI with QUMA's default conditions and uniquely mapped reads were used to calculate DNA methylation level per CpG dinucleotide site. Due to mappability issues, 20 mismatches were allowed for mapping the amplicons over the chimpanzee ZNF396 gDMR locus. When SNPs were available, reads perfectly matching each parental genome were extracted and aligned separately for allele-specific analyses.

**ChIP-sequencing in mouse oocytes.** Germinal vesicle oocytes (GVOs) were isolated from 7–10-weeks-old C57BL/6J females. The zona pellucida was dissolved by passaging the oocytes through an acid Tyrode's solution, followed by neutralization in M2 media. The oocytes were re-suspended in nuclear isolation buffer (Sigma), flash-frozen in liquid nitrogen and stored at −80 °C until usage. RNA polymerase II ChIP-seq libraries were prepared from ~200 GVOs using a modified version of ULI-NChIP-seq[77]. Briefly, chromatin was fragmented with MNase (NEB), diluted in native ChIP buffer (20 mM Tris-HCl pH 8.0; 2 mM EDTA; 150 mM NaCl; 0.1% Triton X-100) containing 1 mM PMSF (Sigma) and EDTA-free protease inhibitor cocktail (Roche), and incubated overnight at 4 °C with 0.1 µg of anti-RNA polymerase II monoclonal antibodies (Abcam ab817–8WG16-) and 5 µl of protein A: protein G 1:1 Dynabeads (Thermofisher). Antibody-bond chromatin was washed four times in low salt buffer (20 mM Tris-HCl pH 8.0; 2 mM EDTA; 150 mM NaCl; 1% Triton X-100; 0.1% SDS) and eluted at 65 °C for 1 h in elution buffer (0.1 nM NaCO3; 1% SDS). DNA was then extracted using phenol:chloroform and precipitated in 75% ethanol, followed by paired-end library construction. Libraries were sequenced (75 bp paired-end) on a NextSeq 500 according to manufacturer's protocols.

**ERV deletions using CRISPR-Cas9.** The ERVs at *Slc38a4* and *Impact* were deleted in mouse ESCs using transient delivery of sgRNAs and Cas9 from an expression vector similar to pX330-U6-Chimeric_BB-CBh-hSpCas9 (Addgene #42230)[78]. Plasmid phU6-gRNA-CBh-hCas9 was obtained from Rupesh Amin and Mark Groudine and first modified by insertion of the PGK-puro-pA selectable marker from pPGKpuro (Addgene #11349)[79] to obtain pPuro2-hU6-gRNA-CBh-Cas9. Individual sgRNAs were designed from http://crispr.mit.edu/ and incorporated on a single forward oligonucleotide (IDT) including a 5′ SapI site and a 29-bp region of complementarity to a universal reverse primer containing an XbaI site. All primer sequences are shown in Supplementary Data 3. For each sgRNA, 10 pmoles of forward and universal reverse primers were annealed and extended with Q5 high-fidelity DNA polymerase (NEB) according to the recommended conditions and with the following program: 3 min/96 °C, three cycles of 3 s/96 °C-30 s/50 °C-3 min/72 °C, and final extension for 5 min/72 °C. The PCR reaction was cleaned with the QIAquick PCR Purification Kit (QIAGEN), and cloned as a ~110-bp SapI-XbaI fragment into pPuro2-hU6-gRNA-CBh-Cas9. For each deletion, we designed 4 different sgRNAs, 2 targeting each side of the ERV (Supplementary Fig. 9), and tested the efficiency of each individual guide using an endonuclease assay. For these assays, 1 µg of each sgRNA plasmid were transfected in C57BL/6N C2 ESCs[80] by lipofection (Lipofectamine 2000, ThermoFisher). After 16 h, puromycin selection (4 µg/ml) was started and continued for 48 h, after which the cells were grown without selection for 5 days. Whole cell populations were collected and prepared for PCR by HotSHOT[81]. Genomic PCR was performed with primers flanking the sgRNA site (Supplementary Data 3) and purified amplicons were melted, reannealed, and then digested using T4 Endonuclease I (NEB). Cut/uncut ratios were calculated following agarose gel electrophoresis and the most efficient sgRNA pairs were chosen for subsequent use. For each deletion, sgRNA plasmid pairs (total of ~10 µg each) were electroporated into C2 cells as previously described[80]. After 48 h of puromycin selection, cells were grown in ESM for 7 days. Single colonies were picked and expanded, then screened for full-length LTR deletion by PCR, Sanger sequencing, and chromosome contents. Euploid lines containing full-length deletion were selected for aggregation chimera production.

**Injection chimera production.** Blastocysts were collected at E3.5 from female albino C57Bl/6J-TyrC2J mice following natural mating, and placed in a 200 ul drop of KSOM overlaid with embryo-tested mineral oil and incubated at 37 °C until ready for injection. Pooled clones of *Slc38a4* LTR KO (#1, #2 and #10) or *Impact* LTR KO (#3 and #5) ESCs were suspended in injection media (DMEM with HEPES + 25% KO ES medium without LIF/2i), and blastocysts were injected with 8–10 ES cells per embryo. After injection, 10–20 embryos were implanted into E2.5 pseudo pregnant mice via uterine transfer. In total, 28 and 7 chimeric pups were born for the *Slc38a4*[MT2AKO] and *Impact*[MTCKO] mutant ESCs, respectively. From these, at least three chimeric males per line transmitted a deletion allele. Transgenic mice were bred and maintained in the Centre for Disease Modelling, Life Sciences Institute, University of British Columbia, under pathogen-free conditions. In all heterozygous genotypes, the maternally inherited allele is always presented first.

**Allele-specific expression analysis.** E13.5 embryos and placentae were dissected from reciprocal F1 crosses between heterozygous LTR KO animals on the C57BL/6 background and WT CAST mice. Embryonic heads for the *Impact* MTCKO and placentae for the *Slc38a4* MT2AKO crosses were used for RNA extraction using Trizol (ThermoFisher). RNA was DNase I-treated (Fisher) at 37 °C for 1 h, then heat inactivated at 65 °C for 15 min. cDNA was synthesized with MMLV-RT (ThermoFisher) using N15 oligonucleotides. RT-PCR was performed with primers described in Supplementary Data 3, and purified PCR products were subjected to restriction enzyme digestion (PvuII for *Slc38a4* or MluCI for *Impact*) and resolved on a 2% agarose gel. Uncut PCR products were sent for Sanger sequencing, and.ab1 files were analyzed using the PHRED software (http://www.phrap.org/phredphrapconsed.html).

**Bisulfite sequencing.** E13.5 embryos and placentae were dissected from reciprocal F1 crosses between heterozygous LTR KO animals on the C57BL/6 background and wild-type CAST/EiJ mice obtained from the Jackson Laboratory (stock number 000735). Bodies were minced and gDNA was extracted by proteinase K digestion followed by phenol:chloroform extraction. One hundred nanogram of gDNA was vortexed to shear the DNA, denatured in 0.2 M NaOH, bisulfite converted in 0.225 M NaOH; 0.0125% hydroquinone; 4 M NaHSO3 overnight at 50 °C, cleaned using the Wizard DNA cleanup kit (Promega) and desulfonated in 0.3 M NaOH at 37 °C for 30 min. Oocytes were harvested from females of the appropriate genotype at 21–28 days of age. GVOs were isolated by passing dissected ovaries through a 100 µm filter using a blunt tool, applying the filtrate to a 35 µm filter, then back-washing the trapped GVOs from the filter. GVOs were further purified by manual collection. >200 Oocytes were pooled from multiple females (typically 4–5 females total). Oocyte DNA was harvested by incubation in 0.1% SDS 1 µg/µl Proteinase K in the presence of 1 µg lambda DNA at 37 °C for 60 min followed by 98 °C for 15 min. DNA was converted using EZ DNA Methylation Gold Kit (Zymo) per manufacturer's protocols. Three biological replicates for oocytes and three independent BS conversions for embryos were amplified using semi-nested primers (Supplementary Data 3) and touchdown-PCR conditions. Purified PCR products were TA-cloned into pGEM-T (Promega), and sequenced at the McGill/Genome Quebec Innovation Centre. Sequences were analyzed using BiQ Software (https://biq-analyzer.bioinf.mpi-inf.mpg.de/). Informative SNPs were identified in final sequences and used to identify individual C57BL/6 or CAST strands.

**Quantitative RT-PCR.** To measure total levels of expression, quantitative PCR (RT-qPCR) was performed on cDNA (described above) using Eva Green (Biotium) on a Step-One Plus Real time PCR system (Applied Biosystems). All reactions were run as follows: 2 min at 95 °C, then 40 cycles of 30 s at 95 °C, 30 s at 60 °C, 30 s at 72 °C, and fluorescence was read at 80 °C. Ct values of six biological replicates from two litters for each stage were averaged and used to calculate relative amounts of transcripts, normalized to levels of the housekeeping gene *Ppia*. Primer sequences are available in Supplementary Data 3.

**Reporting summary.** Further information on research design is available in the Nature Research Reporting Summary linked to this article.

## Data availability
Data generated for this manuscript were deposited at GEO datasets under the accession GSE126363. Tracks for oocyte data analyzed here can be accessed through the data hub https://datahub-d85hei26.udes.genap.ca/NatComm2019/hub.txt. Datasets analyzed for this manuscript are detailed in Supplementary Data 1. The source data underlying Figs. 3c, d, 5d–g, and 6d–g and Supplementary Figs 6 and 9a, b, f, g are provided as a Source Data file.

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

## Acknowledgements

We thank Rupesh Amin and Mark Groudine for their CRISPR-Cas9 vector, Hiromi Kamura, Prof. Kenichiro Hata (National Center for Child Health and Development), Tomoya Takashima, Dr. Keisuke Tanaka, and Prof. Tomohiro Kono (Tokyo University of Agriculture) for their technical assistance with targeted bisulfite sequencing analysis, Julien Richard Albert and Amanda Ha for technical assistance, and Dixie Mager and Carolyn Brown for helpful discussions. M.C.L. is supported by CIHR Grant PJT-153049 and NSERC Discovery Grant 2015-05228. L.L is supported by CIHR Grant MOP-119357 and NSERC Discovery Grant 386979-12. H.K. is supported by MEXT KAKENHI Grant Numbers 15H05579 and JP18H05553. H.I is supported by MEXT KAKENHI Grant Numbers 15H05242 and 18H04005. A portion of primate samples were provided through the Great Ape Information Network (supported by The Cooperative Research Programs of the Primate Research Institute and Wildlife Research Center, Kyoto University). H.K. and H.I. are supported by The Cooperative Research Programs of the Genome Research for BioResource, NODAI Genome Research Center, Tokyo University of Agriculture, and Primate Research Institute and Wildlife Research Center, Kyoto University.

## Author contributions

L.L., M.C.L., H.K., J.B., and A.B.B. conceived the project and designed the experiments. L.L., A.B.B., K.N.J., and J.B. performed murine experiments. H.K., K.N., and H.I. performed primate experiments. A.B.B., J.B., and H.K. performed data analyses. L.L., M.C.L., H.K., J.B., and A.B.B. wrote the manuscript.

## Competing interests

The authors declare no competing interests.
