## [Peer Review File · Nature Communications]

REVIEWERS' COMMENTS:

Reviewer #1 (Remarks to the Author):

The authors have sufficiently addressed my minor concerns regarding the expression effects of SLC38a4 and Impact after LTR deletion, and included critical genetic evaluation of transmission of these deletion alleles. I commend them on their work and I recommend publication.

I still disagree with the author's interpretation of how substantially novel the LIT-mediated DNA methylation deposition through transcription in oocytes mechanism is from previous TE-directed cis and trans epigenetic regulation. While in this work they focus on oocyte DNA methylation, I'm sure the authors are aware that in the male mammalian germline piRNAs derived from piRNA clusters containing TE sequence homology need to be transcriptionally activated to mediate DNA methylation deposition, both in cis at piRNA generating loci and in trans elsewhere in the genome. This LIT-directed DNA methylation deposition mechanism is also analogous to the RNA-induced transcriptional silencing (RITS) first described in pombe--which although doesn't couple transcription to DNA methylation (which is absent from the pombe genome), does couple transcription to silencing through H3K9me deposition in cis.

In this reviewer's estimation, this is another variation on upon a theme--but I think their approach of addressing it head-on in the discussion will continue to spark a lively debate in the TE field.

Reviewer #2 (Remarks to the Author):

I thank the authors for carefully addressing comments I made on their manuscript (reviewed previously for Nature Genetics). The writing of the manuscript is much improved and many points clarified. By looking at a few known igDMRs, the authors convincingly demonstrated that expression of a ERV in oocytes leads to methylation of a downstream igDMR, which is maintained in placenta cells as a maternally imprinted igDMR. This is a novel mechanism for the evolution of imprinting.

I'm overall happy with the revised manuscript and support its publication by Nature Communications. I have a few minor comments.

1. The study still lacks statistical rigor. I suggest the authors provide statistical significance assessment for all their observations/conclusions. For example, "the presence of a lineage-specific proximal LTR that initiates a transcript in oocytes that overlaps with a genic gDMR is associated exclusively with genes showing evidence of a species-specific igDMR" – how different is it from expectation? Another example, "For 11/15 and 7/11 syntenic putative igDMRs in chimp and in macaque, the presence of an LTR correlates with the methylation status" – what is the expectation?

2. The manuscript tuned down some of the claims, however, there are still many places where the authors making claims without the actual data. For example, in several places the authors used bimodal distribution of methylation to indicate imprinting ("shows a clear bimodal distribution of hypermethylated and hypomethylated reads in the placentae of both species, indicative of conservation of LIT-associated HECW1 imprinting in Catarrhines", "shows a bimodal distribution of hypermethylated and hypomethylated sequenced reads in the placenta (Fig. 2e and Supplementary Fig. 6a), indicative of conserved placental-specific ST8SIA1 imprinting within the Hominoidea"). Bimodal distribution is consistent with imprinting, but many other reasons can result in bimodal distribution. Bimodal distribution does NOT indicate imprinting.

3. Another example is when the authors rejected the suggestion to do 5' RACE citing lack of material in at least two places. I think that's fine. However without the data there are claims that one must be cautious to make. You can make a prediction, but you cannot use predicted results as support of your claim. Just be careful with what you can or cannot claim when the actual data is not available.

Reviewer #1 (Remarks to the Author):

The authors have sufficiently addressed my minor concerns regarding the expression effects of SLC38a4 and Impact after LTR deletion, and included critical genetic evaluation of transmission of these deletion alleles. I commend them on their work and I recommend publication.

We thank the reviewer for their positive comments.

I still disagree with the author's interpretation of how substantially novel the LIT-mediated DNA methylation deposition through transcription in oocytes mechanism is from previous TE-directed cis and trans epigenetic regulation. While in this work they focus on oocyte DNA methylation, I'm sure the authors are aware that in the male mammalian germline piRNAs derived from piRNA clusters containing TE sequence homology need to be transcriptionally activated to mediate DNA methylation deposition, both in cis at piRNA generating loci and in trans elsewhere in the genome. This LIT-directed DNA methylation deposition mechanism is also analogous to the RNA-induced transcriptional silencing (RITS) first described in pombe--which although doesn't couple transcription to DNA methylation (which is absent from the pombe genome), does couple transcription to silencing through H3K9me deposition in cis.

We have expanded our discussion to include the distinctions between the piRNA-mediated germline DNAm and our reported LIT-directed DNAm. LIT-directed DNAm differs from piRNA-induced DNAm in several fundamental ways. First, piRNA-induced DNAm is not a cis-acting mechanism, but rather is targeted to the genome via the piRNA-induced Silencing Complex following production and processing of piRNAs, loading into complexes, and retargeting of homologous nascent RNA or DNA sequences, which may be located anywhere in the genome. This is distinct from the transcription-coupled DNAm we report here, as the ERVs responsible for promoting transcription at our loci of interest do not gain DNAm; rather, sequences downstream of the ERVs (ie in cis), which are not necessarily repetitive themselves, gain DNAm as a consequence of these novel promoters active in the germline. Second, RNA itself does not play a role in this gain of DNAm, as outlined above for both the piRNA-induced and RNA-induced transcriptional silencing. Rather, LIT-directed DNAm is coupled to active RNA PolII via recruitment of Setd2 and in turn, deposition of H3K36me3. The RNA produced by this transcription is inconsequential to the DNAm laid down in its wake, unlike the other models proposed.

Nevertheless, to address this comment, we have added the following sentence to the end of our Discussion:

“In contrast, *de novo* DNA methylation at the *Rasgrf1* locus in male germ cells involves a *trans* mechanism, whereby an RMER4B LTR at the 3'-end of the gDMR is targeted for DNA methylation by a yet to be fully characterized nuclear piRNA pathway active in prospermatogonia.”

In this reviewer's estimation, this is another variation on upon a theme--but I think their approach of addressing it head-on in the discussion will continue to spark a lively debate in the TE field.

As discussed above, we believe that the phenomenon we describe is more than a “variation on upon a theme”. Given that our manuscript is already at the limit in terms of length, we believe that a detailed discussion of the different mechanisms by which TEs influence DNA methylation is more appropriate for a review article on the topic.

Reviewer #2 (Remarks to the Author):

I thank the authors for carefully addressing comments I made on their manuscript (reviewed previously for Nature Genetics). The writing of the manuscript is much improved and many points clarified. By looking at a few known igDMRs, the authors convincingly demonstrated that expression of a ERV in oocytes leads to methylation of a downstream igDMR, which is maintained in placenta cells as a maternally imprinted igDMR. This is a novel mechanism for the evolution of imprinting.

I'm overall happy with the revised manuscript and support its publication by Nature Communications. I have a few minor comments.

We thank the reviewer for their positive comments.

1. The study still lacks statistical rigor. I suggest the authors provide statistical significance assessment for all their observations/conclusions. For example, “the presence of a lineage-specific proximal LTR that initiates a transcript in oocytes that overlaps with a genic gDMR is associated exclusively with genes showing evidence of a species-specific igDMR” – how different is it from expectation? Another example, “For 11/15 and 7/11 syntenic putative igDMRs in chimp and in macaque, the presence of an LTR correlates with the methylation status” – what is the expectation?

Our previous publication (Brind'Amour et al. 2018) reported the prevalence of LITs in mouse, rat, and human, and showed how many CGIs are methylated as a consequence. We had not, however, directly compared the numbers obtained in that publication with those observed here. As requested by the reviewer, we have added the following text to our Results section (line 96): “Compared with all CGIs genome-wide, this represents a significant enrichment of LITs at igDMRs (mouse: 4/21 igDMRs vs 152/16023 CGIs, chi-square $p = 1.17 \times 10^{-17}$; human: 17/125 igDMRs vs 70/31144 CGIs, chi-square $p = 7.42 \times 10^{-219}$).”

For the evolution of an igDMR, several criteria must be fulfilled, including establishment of differential DNAm in the gametes and maintenance of the differential methylation post-fertilization. The acquisition of a novel oocyte-specific promoter through retrotransposition of an ERV produces a block of DNAm that could potentially lead to an imprint, but it must overlap a CGI that is competent to maintain that DNAm through development and doesn't see a gain of DNAm through other means. If a CGI is differentially methylated but gains DNAm through development, the imprint is lost as the alleles become epigenetically equivalent. This is seen at a handful of genes, and for one of our genes of interest this happens in some tissue types (*retro-Coro1c*, in the thymus for example). In other words, the “expectation” of how many LTRs drive the establishment of igDMRs goes beyond the simple statistical probability of finding an active LTR promoter upstream of a CGI.

2. The manuscript tuned down some of the claims, however, there are still many places where the authors making claims without the actual data. For example, in several places the authors used bimodal distribution of methylation to indicate imprinting (“shows a clear bimodal distribution of hypermethylated and hypomethylated reads in the placentae of both species, indicative of conservation of LIT-associated HECW1 imprinting in Catarrhines”, “shows a bimodal distribution of hypermethylated and hypomethylated sequenced reads in the placenta (Fig. 2e and Supplementary Fig. 6a), indicative of conserved placental-specific ST8SIA1 imprinting within the Hominoidea”). Bimodal distribution is consistent with imprinting, but many other reasons can result in bimodal distribution. Bimodal distribution does NOT indicate imprinting.

In the case of bimodal distribution of DNAm, we agree that this datapoint in and of itself does not necessarily indicate imprinting, though in our view it is the most likely explanation. Regardless, additional independent evidence supports our conjecture that these loci are imprinted:

1. Each of these CGIs show differential methylation between the parental genomes in germ cells (Figure 1b, Supplemental Figure 4a)
2. DNAm is lost in *Setd2-KO* Oocytes (Supplemental Figure 1b)
3. Loci for which we are able to obtain allelic DNAm data show clear maternal DNAm
4. Each of these loci have already been shown to be imprinted in human

As these data are consistent with our model, we feel comfortable suggesting that the loci for which we do not have allelic DNAm data are likely imprinted.

3. Another example is when the authors rejected the suggestion to do 5' RACE citing lack of material in at least two places. I think that's fine. However without the data there are claims that one must be cautious to make. You can make a prediction, but you cannot use predicted results as support of your claim. Just be careful with what you can or cannot claim when the actual data is not available.

We don't believe that our claims are overstated. The reviewer previously suggested 5'RACE experiments in oocytes to show promoter usage in the case of LITs, and in offspring of LTR KO animals to show promoter usage when loss of imprinting is observed. In the case of LITs, we do not claim that transcription initiation is not taking place in oogenesis from the canonical genic promoters (exon 1) of genes overlapped by a LIT (although in most cases the RNAseq data suggests very little activity); rather, we show that the use of the LTR itself as a promoter, agnostic to other promoter usage, is the critical feature of LITs in the establishment of DNAm at the CGIs investigated. Likewise, we make no claims about the promoter usage at *Impact* and *Slc38a4* in our LTR KO maternal transmission animals. We simply report the loss of DNAm at the igDMR and the concomitant loss of imprinting at the RNA level. While it is possible that an alternative promoter becomes active on the maternal allele in the absence of the LTR, this phenomenon is really not relevant to the main focus of our paper, which is the critical role that upstream LTRs play in the establishment of DNAm at these igDMRs and in turn genomic imprinting.